# Fast and Excellent Enhanced Photocatalytic Degradation of Methylene Blue Using Silver-Doped Zinc Oxide Submicron Structures under Blue Laser Irradiation

Samer H. Zyoud [1,2,3,4,*], Ibrahim S. Yahia [2,5,6,7], Moyad Shahwan [3,8], Ahed H. Zyoud [9], Heba Y. Zahran [5,6,7], Mohamed Sh. Abdel-wahab [10], Malek G. Daher [11], Mohamed Nasor [3,12], Ghaseb N. Makhadmeh [13], Nageeb Hassan [3,8], Akram Ashames [3,8] and Naser Qamhieh [14]

1    Department of Mathematics and Sciences, Ajman University, Ajman P.O. Box 346, United Arab Emirates
2    Nonlinear Dynamics Research Center (NDRC), Ajman University, Ajman P.O. Box 346, United Arab Emirates
3    Center of Medical and Bio-Allied Health Sciences Research (CMBHSR), Ajman University, Ajman P.O. Box 346, United Arab Emirates
4    School of Physics, Universiti Sains Malaysia (USM), Penang 11800, Malaysia
5    Laboratory of Nano-Smart Materials for Science and Technology (LNSMST), Department of Physics, Faculty of Science, King Khalid University, Abha P.O. Box 9004, Saudi Arabia
6    Research Center for Advanced Materials Science (RCAMS), King Khalid University, P.O. Box 9004, Abha 61413, Saudi Arabia
7    Nanoscience Laboratory for Environmental and Biomedical Applications (NLEBA), Semiconductor Lab., Metallurgical Lab. 1., Department of Physics, Faculty of Education, Ain Shams University, Cairo 11566, Egypt
8    Department of Clinical Sciences, College of Pharmacy and Health Sciences, Ajman University, Ajman P.O. Box 346, United Arab Emirates
9    Department of Chemistry, SSERL, An-Najah National University, Nablus P.O. Box 7, Palestine
10   Materials Science and Nanotechnology Department, Faculty of Postgraduate Studies for Advanced Sciences, Beni-Suef University, Beni-Suef 62511, Egypt
11   Physics Department, Islamic University of Gaza, Gaza P.O. Box 108, Palestine
12   Department of Biomedical Engineering, Ajman University, Ajman P.O. Box 346, United Arab Emirates
13   Department of Physics, Bio-Medical Physics Laboratory, Jordan University of Science and Technology, P.O. Box 3030, Irbid 22110, Jordan
14   Department of Physics, United Arab Emirates University, Al-Ain P.O. Box 15551, United Arab Emirates
*    Correspondence: s.zyoud@ajman.ac.ae

**Abstract:** In this study, laser-assisted chemical bath synthesis (LACBS) was used to prepare pure and Ag-doped ZnO submicron structures using a simplified hydrothermal approach that did not require a catalyst. The photocatalytic degradation of Methylene Blue was investigated under blue laser irradiation ($\lambda$ = 444.5 nm and I = 8000 lx). The doping concentration varied (2%, 4%, 6%, 8%, tando 10%) and was prepared by LACBS using a continuous blue laser (P = 7 W, $\lambda$ = 444.5 nm) for the first time. XRD, FE-SEM, EDX, and UV-Vis investigated the characteristics of the samples produced by the LACBS. ZnO: $Ag_{(10\%)}$ submicron flowers are essential in rapid photodegradation under blue laser irradiation. The high surface area and catalytic activity of the prepared Ag-decorated ZnO are attributed to this improved photocatalytic activity. Using UV-visible spectroscopy, the photocatalytic efficiency was determined from the absorption spectra. The separation of photo-generated electron-hole pairs was facilitated, and the absorption edge of the hybrid submicron structures shifted into the visible spectrum region due to a combination of the Ag plasmonic effect and surface imperfections in ZnO. Effective visible light absorption was achieved via band-edge tuning, which increased the ZnO:Ag submicron structures' ability to degrade dyes.

**Keywords:** LACBS; pure and Ag-doped ZnO; methylene blue; photocatalytic; submicron structures

## 1. Introduction

Chemicals with a carbon base and persistent organic pollutants are resistant to environmental deterioration and may not be eliminated by treatment procedures. Their persistence

may have detrimental effects on animals and people's health. Due to its significant potential as a green and eco-friendly method of removing persistent organic contaminants to promote clean water security, the solar photocatalysis process has drawn increasing interest [1]. Due to global water scarcity brought on by climate change and ineffective water resource management, water reclamation and reuse have recently attracted more attention (i.e., restricted clean water and water demands surpassing available supplies). A developing global economy and an increasing population of countries make access to clean water a growing challenge [2]. Implementing wastewater reclamation and reuse initiatives to promote sustainable water development and management is one of the most alluring responses to water concerns. However, there are still worries that cleaned water may still contain persistent organic contaminants. Adsorption, membrane separation, and coagulation are three water treatment methods used to remove persistent organic pollutants from water streams; however, these methods merely concentrate or convert the refractory organic pollutants from the water to the solid phase [3]. The secondary contaminants must be treated, which will cost more money, and the adsorbents must be renewed [4]. Improved oxidation techniques have been suggested to eliminate refractory organic pollutants, especially those with low biodegradability. Advanced oxidation techniques have several benefits, including quick degradation rates, the mineralization of organic compounds into environmentally friendly products, the capacity to work at room temperature and pressure, and a decrease in the toxicity of organic compounds [5].

In the photocatalytic degradation of dyes, many systems have been employed with varying degrees of effectiveness. However, semiconductors are among the most investigated materials and are used in various ways, including on their own, with defects, doped, or in combination with other materials. Due to its superior mechanical, thermal, and optoelectronic properties, ZnO has been one of the most versatile materials studied in recent years, whether in bulk, nanostructures, or epilayers. [6]. ZnO is a biocompatible, safe material with a broadband gap semiconductor (3.37 eV) and a high excitation binding energy (~60 meV) [7]. ZnO nanostructure is a promising material for various technological applications, such as solar cells, photodetectors, gas sensors, energy, photodetectors, and photoconductive devices [8]. ZnO hierarchical nanostructures, such as nanotubes, nanowires, and nanorods, are successfully generated, in addition to nanosized particles, using particular synthesis methods and under appropriate reaction conditions [9]. ZnO nanostructures have many uses, but photocatalysis is the most promising [10]. Organic dye molecules can be efficiently degraded by the good oxidizing agents in ZnO, which belong to the photo-generated reactive oxygen species. Doping semiconductor metal oxides with metals (such as Al, Ni, Mn, Mg, Au, Ag, etc.) is the most efficient way to improve their performance or tailor their activity to particular wavelength ranges in photocatalysis [11]. On the ZnO surface, metallic dopants provide new energy levels that can receive photo-generated electrons from the conduction band and significantly lessen the effects of recombination [12].

By utilizing ZnO and $TiO_2$ under sun irradiation, Fenoll et al. [13] compared the photodegradation of fungicides in leaching water. They discovered nonstoichiometric ZnO, which outperforms $TiO_2$ as a photocatalyst when exposed to sun radiation. $La^{3+}$, $Nd^{3+}$, and $Sm^{3+}$ doping were utilized by Khatamian et al. [14] to enhance ZnO's photoactivity in the breakdown of 4-nitrophenol. Sin et al. [15] significantly improved the ZnO photocatalytic degradation efficiency of phenol under natural sunlight by doping ZnO with $Eu^{3+}$ ions. According to Jiang et al., doping $TiO_2$ with silver led to 100% photodegradation after being exposed to light for two hours [16]. Albiter et al. prepared Ag-doped $TiO_2$ that degraded 85% dye after 1 hr [17]. After 150 minutes of radiation exposure, Avciata et al. prepared $TiO_2$ doped with Ag nanoparticles, which accelerated dye degradation to 75% [18]. To increase the photoactivity of ZnO in the degradation of 4-nitrophenol, Khatamian, et al. [14] employed $La^{3+}$, $Nd^{3+}$, and $Sm^{3+}$ doping.

Advanced laser technology and a better understanding of laser interactions with the materials have been beneficial for rapid treatment and designing novel fabrication

techniques that can produce functional nanomaterials. Laser-assisted systems work in continuous wave or pulse mode with UV to IR range [19]. The application of lasers can induce controlled chemical reactions, heating effects, and other complex phenomena related to nucleation and particle growth in synthesizing metal oxide nanostructures. The benefits of LACBS include its simplicity, low cost, speed, high yield, and scalability [20]. The development of LACBS technology has enabled the production of nanostructures with various morphologies, which is essential for developing sensing systems. This method's basis is the photo-thermal effect brought about by a focused laser, which produces a contained temperature field at the required location with excellent control [21]. There is meticulous control over the fabrication parameters and physical properties, and the performance of metal oxides synthesized by LACBS shows significant enhancement [22]. This study examined undoped and LACBS-fabricated Ag-doped ZnO nanostructures. In fact, for the first time, the photocatalytic degradation of methylene blue was investigated using nanopowders under blue laser irradiation. In this article, prepared powders were stacked to form nanorods and nanoflowers with a high surface-to-volume ratio and crystallinity, which aids photodegradation. The photocatalytic degradation of MB blue laser irradiation was investigated. This study aimed to determine how different experimental conditions, such as light source, exposure time, and Ag-doped ZnO, affected the results. ZnO:Ag nanoflowers are critical for rapid photodegradation in the presence of blue laser irradiation.

## 2. Methodology

### 2.1. Starting Materials

$[Zn(CH_3COO)_2.(H_2O)_2(\geq 99.99\%)]$ zinc acetate dihydrate, and hexamethylenetetramine $[C_6H_{12}N_4, (99.9\%)]$ were acquired from Sigma Aldrich and utilized exactly as obtained with no further purification. Deionized water was used as the deposition solvent to prepare the precursor solutions. Silver nitrate $(AgNO_3)$ was purchased from Pub-Chem, United States. Methylene Blue (MB) $[C_{16}H_{18}CIN_3C$, (Merck, United States)] dye solution was used as a pollutant example wastewater. The chemical structure of MB is presented in Figure 1.

**Figure 1.** Chemical structure of methylene blue (MB).

### 2.2. Catalyst Preparation

Reagent preparation for zinc oxide (ZnO, 0.1 M): zinc acetate dihydrate $(ZnC_4H_6O4)$ as a solution A: 2.195 g of $ZnC_4H_6O_4$ was dissolved in 50 mL deionized water, stirred well, and sonicated to make it homogeneous. Hexamethylenetetramine $(C_6H_{12}N_4)$ as a solution B: 1.402 g of $C_6H_{12}N_4$ was dissolved in 50 mL deionized water, stirred well, and sonicated to make it homogeneous. Metal Salt solutions preparation (0.1 M): Silver nitrate $(AgNO_3)$ as a solution C: 1.699 g of $AgNO_3$ was dissolved in 50 mL deionized water, stirred well, and sonicated to make it homogeneous.

### 2.3. Synthesis ZnO Submicron Powder

The LACBD technique outlined below produced pure ZnO submicron powder. After 15 min of magnetic stirring at room temperature, the precursor solutions (A and B) dissolved separately. The B solution was then added dropwise to the A solution with continuous stirring at room temperature for 30 min. The resulting solution was magnetically swirled for 120 min at 60 °C with a continuous wave semiconductor blue laser (P = 7 W, λ = 444.5 nm) applied vertically on the mixed solution throughout the process to obtain an evenly growing solution. The precipitate was filtered out, and ethanol was used to rinse it three times. After filtration, the precipitated ZnO nanostructures were left to dry for 30 min at 100 °C. To

ensure total decomposition and the complete removal of all organic materials, the collected powder was subsequently annealed at 400 °C for 4 h.

### 2.4. Synthesis Ag-Doped ZnO Submicronflowers

The following procedures were conducted to synthesize Ag-doped ZnO submicron structures by LACBS. The solutions A, B, and D were magnetically agitated at room temperature for 15 minutes to dissolve independently. The same approach was adopted for synthesizing Ag-doped ZnO by taking an appropriate amount of $AgNO_3$. For the synthesis of 2%, 4%, 6%, 8%, and 10% Ag-doped ZnO, the same procedure was adopted as in Section 2.3. Figure 2 represents a schematic diagram illustrating the working principle for synthesizing pure and Ag-doped ZnO submicron structures.

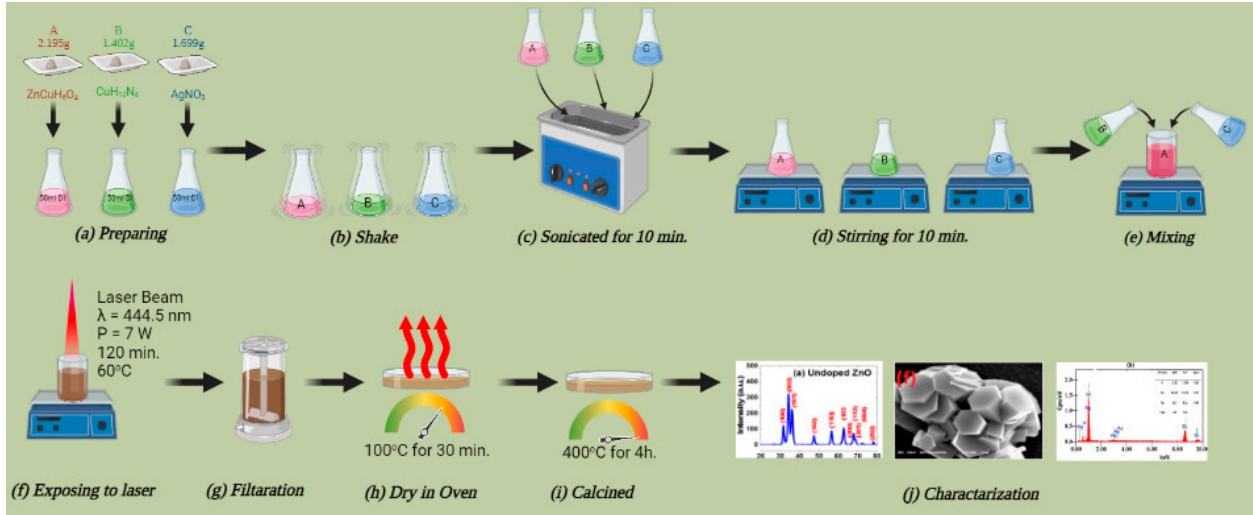

**Figure 2.** Schematic diagram and illustration of the working principle for synthesizing pure and Ag-doped ZnO submicron structures.

### 2.5. Characterization

The crystal characteristics of the produced samples were assessed using an X-ray diffractometer (XRD; Rigaku Japan) for phase and structure identification. The surface morphology and chemical composition of pure and Ag-doped ZnO nanostructures were observed and examined using energy-dispersive X-ray (EDX) equipment and field emission scanning electron microscopy (FE-SEM, JEOL JSM-7600F) with an accelerating voltage of 30 KV. UV-Vis spectrometer model, PE lambda 750S was used to evaluate the optical absorption of materials. It was necessary to confirm the optical absorption spectra of this solution using a UV-Vis spectrum analyzer. Fourier-transform infrared spectra (FTIR) were employed to comprehend the chemical molecule.

### 2.6. Photocatalyst Experiment

#### 2.6.1. Preparation of Methyl Blue (MB) Solution

20 ppm of Methylene blue was prepared by dissolving 0.1 g in 10 mL of the flask and sonicating well to make a firmly homogeneous test solution. Pour the above 10 mL of 0.1% solution into 500 mL and make up the solution. Sonicate before each test to ensure homogeneous dissolution and 20ppm distribution of dye molecules.

#### 2.6.2. Photocatalytic Degradation of MB

The photocatalytic degradation of MB was investigated at room temperature under blue laser light irradiation ($\lambda = 444.5$ nm) to eliminate any thermal catalytic influence. The MB solution (20 ppm) was added to a precise quantity (10 mg) of manufactured catalysts, including pure ZnO, $ZnO:Ag_{(2\%)}$, $ZnO:Ag_{(4\%)}$, $ZnO:Ag_{(6\%)}$, $ZnO:Ag_{(8\%)}$, and $ZnO:Ag_{(10\%)}$.

It was then continuously magnetically stirred. Water, MB, and catalysts were agitated for 15 minutes in complete darkness to achieve adsorption-desorption symmetry. The responded materials were later exposed to a blue laser to research photodegradation. At regular intervals of five minutes during the irradiation operations, a specific volume of the liquid (5 mL) was removed. The optical absorption spectra of the removed samples were checked using a UV/Vis spectrum analyzer at 656.4 nm. The degree of degradation of dyes, rate of degradation ($K_{app}$), and photodegradation efficiency (PDE) was calculated using the following equations:

$$\frac{C}{C_o} = \frac{A}{A_o} \tag{1}$$

$$ln\frac{C_o}{C} = k_{app}t \tag{2}$$

$$PDE\% = \left(1 - \frac{C}{C_o}\right) \times 100\% \tag{3}$$

where $C_o$ represents the initial concentration of MB and $C$ is the MB concentration after irradiation with light. Figure 3 illustrates the schematic diagram and illustration of the photocatalytic study.

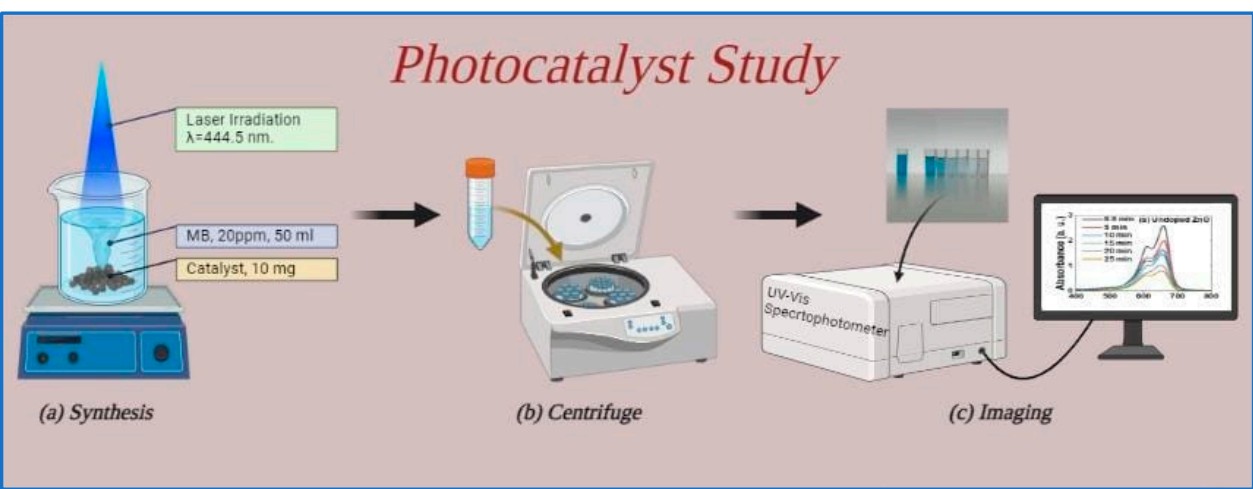

**Figure 3.** Schematic diagram and illustration of the photocatalyst study.

### 3. Results and Discussion

The samples of pure ZnO and Ag-doped ZnO, namely [pure ZnO, ZnO:Ag$_{(2\%)}$, ZnO:Ag$_{(4\%)}$, ZnO:Ag$_{(6\%)}$, ZnO:Ag$_{(8\%)}$, and ZnO:Ag$_{(10\%)}$] submicron structures were obtained via LACBS using a continuous-wave semiconductor laser (P = 7 W, λ = 444.5 nm). The structural, morphological, and optical properties of samples were examined.

### 3.1. X-ray Diffraction (XRD)

Figure 4 represents the typical XRD pattern of the samples of pure ZnO, ZnO:Ag$_{(2\%)}$, ZnO:Ag$_{(4\%)}$, ZnO:Ag$_{(6\%)}$, ZnO:Ag$_{(8\%)}$, and ZnO:Ag$_{(10\%)}$. In the case of pure ZnO, there are distinct diffraction peaks representing (100), (002), (101), (102), (110), (102), (110), (103), (220), (221) and (202) lattice planes that confirm the wurtzite structure, which agrees with the JCPDS card no.01-075-0576 [23]. The sample had highly intense characteristic peaks at 2θ = 31.63°, 34.26°, and 36.04° corresponding to the lattice planes (100), (002), and (101), respectively. The characteristic diffractions of high intensity and the XRD pattern devoid of any other impurity peaks indicated ZnO crystalline and single-phase hexagonal wurtzite crystals. Moreover, (002) lattice plane orientation was strongly preferred for the sample, showing that the c-axis favored nanostructure growth.

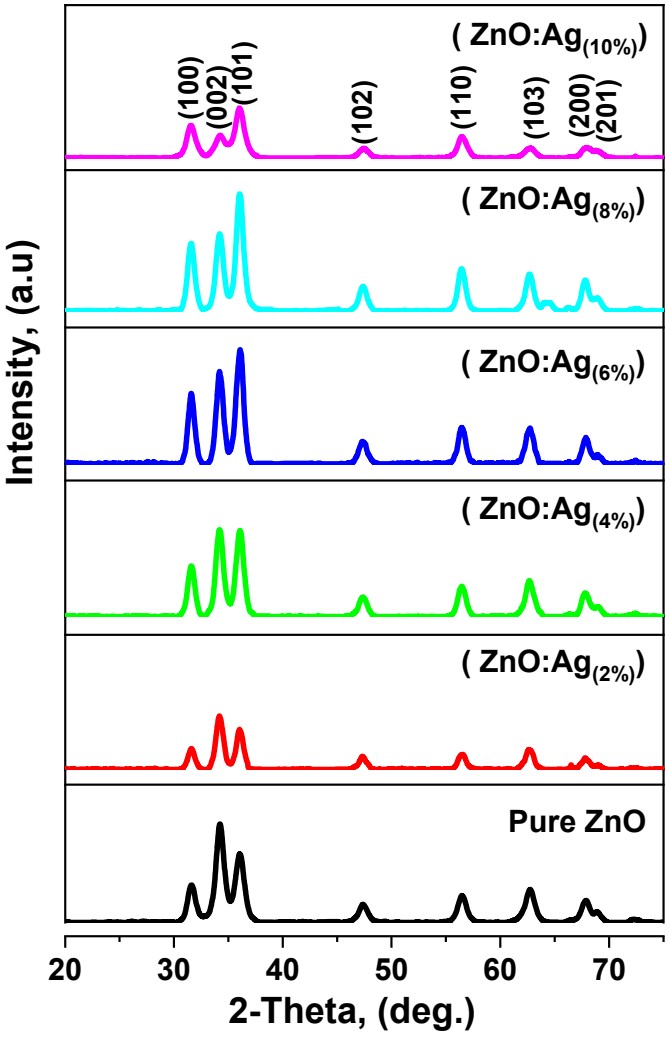

**Figure 4.** XRD spectra of samples undoped and Ag doped ZnO synthesized by LACBS.

The XRD patterns of the ZnO:Ag submicron structure shows intense and narrow peaks indicating good crystallinity of the sample. However, the intensity of the characteristic ZnO diffractions was reduced with a negligible shift in diffraction angle, signifying a decrease in crystallinity. The XRD pattern of ZnO:Ag was observed with a little shifting in corresponding and changing in peak form. No peaks related to silver oxide were found. The ionic radii of $Ag^+$ (1.22 Å) and $Zn^{2+}$ (0.72 Å) designated that the substitution of $Zn^{2+}$ ion by $Ag^+$ ion into the ZnO lattice was implausible [24]. So, the crystalline Ag phase was formed, and those Ag agglomerates may remain somewhere, typically stuck on the ZnO surface [Ag-ZnO]. The diffraction peak corresponding to the (002) plane was observed for ZnO:Ag$_{(2\%)}$ in the highest intensity, indicating the preferential growth of the nanostructure along the c-axis. The diffraction peaks corresponding to (002) and (101) planes were observed for ZnO:Ag$_{(4\%)}$ in the highest intensity, indicating the preferential growth of the nanostructure along the c-axis and b-axis. The diffraction peak corresponding to the (101) plane was observed for ZnO:Ag $_{(6\%)}$ and ZnO:Ag$_{(8\%)}$ in the highest intensity sharp, indicating high crystallinity of the sample and the preferential growth of the nanostructure along the b-axis. Hexagonal wurtzite structure. The reduction in the intensity of the traditional ZnO diffractions implies that ZnO:Ag$_{(10\%)}$ decreased the crystallinity of the powder in comparison with the pure sample. This could result in vanishing (002) lattice diffractions in the XRD pattern. For ZnO:Ag$_{(10\%)}$, the (100) and (101) lattice planes' diffraction in the highest intensity orientation was highly preferred, indicating that the a-axis and b-axis favored nanostructure growth. This might be explained by the solubility of zinc acetate

being dependent on the type of solvent. In contrast to other surfactants, DI water allows zinc acetate to dissolve more quickly and easily. Solubility is, therefore, crucial in lowering the size of submicron-crystallites. Consequently, chemical reactions began when the solution was exposed to laser irradiation, producing tiny ZnO submicron structures [25].

Table 1 shows the structural parameters of doped and undoped ZnO submicron structures formed by LACBS with varied doping concentrations along diffraction peaks (100), (002), and (101) plans. Parameters included peak position (2θ), intensity, full width at half maximum (FWHM), lattice constants (a, b, and c), and internal strain stress ($\varepsilon_a$, $\varepsilon_c$). There was little visible change in parameters because of the change in doping concentration; however, the diffraction intensity decreased with increased doping concentration. Extrinsic and intrinsic stresses contributed to the overall focus of doped and undoped ZnO submicron structures, whereas defects and impurities caused inherent stress. Compared to other approaches, XRD was frequently used to evaluate the residual stress of doped and undoped ZnO submicron structures since the strain could be estimated immediately using Bragg's equation. Table 1 shows the computed focus of doped and undoped ZnO submicron structures. The negative sign of stress denotes compressive stress.

**Table 1.** Lattice parameters and structures of pure and Ag-doped ZnO submicron structures of the mean diffraction peak (100), (002), and (101).

| NO | (h, k, l) | 2 θ (deg) | $d^*$ | β (deg) | I | D (Å) | a = b (Å) | c (Å) | c/a | $\varepsilon_a$ | $\varepsilon_c$ | $d^{**}$ | μ | L (Å) |
|---|---|---|---|---|---|---|---|---|---|---|---|---|---|---|
| | (100) | 31.63 | 2.82 | 0.86 | 83 | 1.67 | 3.26 | 5.65 | 5.65 | 0.48 | 8.68 | 2.82 | 0.36 | 2.04 |
| ZnO | (002) | 34.26 | 2.61 | 0.88 | 218 | 1.64 | 3.02 | 5.23 | 5.23 | −7.02 | 0.55 | 2.61 | 0.36 | 1.88 |
| | (101) | 36.04 | 2.48 | 0.99 | 153 | 1.46 | 2.88 | 4.98 | 4.98 | −11.47 | −4.25 | 2.49 | 0.36 | 1.79 |
| | (100) | 31.62 | 2.82 | 0.67 | 47 | 2.14 | 3.26 | 5.65 | 5.65 | 0.50 | 8.70 | 2.82 | 0.36 | 2.04 |
| ZnO:Ag 2% | (002) | 34.22 | 2.61 | 0.82 | 118 | 1.76 | 3.02 | 5.23 | 5.23 | −6.91 | 0.67 | 2.61 | 0.36 | 1.89 |
| | (101) | 36.08 | 2.48 | 0.75 | 89 | 1.93 | 2.87 | 4.97 | 4.97 | −11.56 | −4.35 | 2.48 | 0.36 | 1.79 |
| | (100) | 31.62 | 2.82 | 0.77 | 112 | 1.87 | 3.26 | 5.65 | 5.65 | 0.51 | 8.71 | 2.82 | 0.36 | 2.04 |
| ZnO:Ag 4% | (002) | 34.21 | 2.61 | 0.81 | 194 | 1.78 | 3.02 | 5.24 | 5.24 | −6.89 | 0.69 | 2.62 | 0.36 | 1.89 |
| | (101) | 36.06 | 2.48 | 0.84 | 191 | 1.73 | 2.87 | 4.97 | 4.97 | −11.53 | −4.32 | 2.48 | 0.36 | 1.88 |
| | (100) | 31.61 | 2.82 | 0.69 | 158 | 2.07 | 3.26 | 5.65 | 5.65 | 0.54 | 8.74 | 2.82 | 0.36 | 2.04 |
| ZnO:Ag 6% | (002) | 34.22 | 2.61 | 0.76 | 205 | 1.88 | 3.02 | 5.23 | 5.23 | −6.92 | 0.66 | 2.61 | 0.36 | 1.89 |
| | (101) | 36.06 | 2.48 | 0.78 | 254 | 1.85 | 2.87 | 4.98 | 4.97 | −11.53 | −4.31 | 2.48 | 0.36 | 1.79 |
| | (100) | 31.59 | 2.82 | 0.73 | 179 | 1.97 | 3.26 | 5.66 | 5.66 | 0.59 | 8.79 | 2.83 | 0.36 | 2.04 |
| ZnO:Ag 8% | (002) | 34.21 | 2.61 | 0.82 | 204 | 1.76 | 3.02 | 5.24 | 5.23 | −6.90 | 0.68 | 2.61 | 0.36 | 1.89 |
| | (101) | 36.06 | 2.48 | 0.77 | 311 | 1.89 | 2.87 | 4.98 | 4.97 | −11.52 | −4.31 | 2.48 | 0.36 | 1.79 |
| | (100) | 31.59 | 2.82 | 0.91 | 73 | 1.57 | 3.26 | 5.66 | 5.66 | 0.59 | 8.79 | 2.83 | 0.36 | 2.04 |
| ZnO:Ag 10% | (002) | 34.31 | 2.61 | 1.18 | 51 | 1.23 | 3.01 | 5.22 | 5.22 | −7.17 | 0.39 | 2.61 | 0.36 | 1.88 |
| | (101) | 36.00 | 2.49 | 0.98 | 112 | 1.48 | 2.87 | 4.98 | 4.98 | −11.37 | −4.15 | 2.49 | 0.36 | 1.80 |

Bragg's Law formula Equation (4) was used to figure out the shift in the diffraction angles at the (100), (002), and (101) plans by calculating the decrease in d-spacing [8]:

$$n\lambda = 2d sin\theta \tag{4}$$

where *n*, *θ*, *λ*, and *d* represented the order of diffraction, diffraction angle, X-ray wavelength, and distance between planes, respectively.

The average crystallite size (D) and the sharpest diffraction peak were calculated using the Debye–Scherer formula [8].

$$D\left(\text{Å}\right) = \frac{k\lambda}{\beta cos\theta} \tag{5}$$

where $k$ is the constant ($k$ = 0.9), $\theta$ is the Bragg diffraction angle, $\lambda$ is the X-ray source wavelength ($\lambda$ = 1.5427), and $\beta$ is FWHM.

More hexamethylenetetramine (HMTA) might have decomposed to $OH^-$ when growth periods were increased, causing more grain expansion. As a result, the additional growing time provided more opportunities to re-join more atoms of $Zn^+$ and $OH^-$ ions to the final product, increasing the crystalline size of ZnO [26].

The lattice constants (a, b, and c) of doped and undoped ZnO were calculated using Bragg's law [27]:

$$a\left(\text{Å}\right) = b\left(\text{Å}\right) = \frac{\lambda}{\sqrt{3}sin\theta} \tag{6}$$

$$c\left(\text{Å}\right) = \sqrt{3}a \tag{7}$$

where $\theta$ is the diffraction peak angle and $\lambda$ is the wavelength of the X-ray source. The values of the lattice parameters (a and c) (Table 1) were almost identical to those reported in the (JCPDS) card for ZnO [27].

Equation (8) may be used to calculate the perpendicular strain ($\varepsilon_a$) of undoped and doped ZnO submicron structures increasing along the a-axis [28]:

$$\varepsilon_a = \frac{a - a_o}{a_o} \times 100\% \tag{8}$$

Equation (9) may be used to calculate the strain ($\varepsilon_c$) of the ZnO TFs formed on the Kapton tape substrate along the c-axis [28]:

$$\varepsilon_c = \frac{c - c_o}{c_o} \times 100\% \tag{9}$$

where $a_o$ and $c_o$ are the standard lattice constants for unstrained ZnO TFs determined from X-ray diffraction pattern data ($a_o$ = 3.2494 Å) and ($c_o$ = 5.2038 ), respectively [8], and the standard lattice constant for unstrained ZnO TFs is (JCPDS card no. 01-079-0206)

The plane d-spacing (d**) is related to the lattice parameters a and c, and the Miller indices (h, k, l) are computed using the theoretical formula. Equation (10) [29]:

$$\frac{1}{d^{2**}} = \frac{4}{3}\left(\frac{h^2 + hk + l^2}{a^2}\right) + \frac{l^2}{c^2} \tag{10}$$

The theoretical Equation (10) yielded the d-spacing values, as well as Braggs' Law Equation (4), which are almost equivalent (Table 1).

The length of the Zn-O bond in ZnO submicron structures was estimated using Equation (11):

$$L = \sqrt{\frac{a^2}{3} + (0.5 - \mu)^2 C^2} \tag{11}$$

where ($\mu$) is the wurtzite structure positional parameter indicating the magnitude of atom movement relative to the next along the c -axis, as given by Equation (12):

$$\mu = \frac{a^2}{3c^2} + 0.25 \tag{12}$$

The computed ZnO bond lengths (Table 1) were quite close to the 1.9767 Å reported in the literature.

### 3.2. Scanning Electron Microscope (SEM)

SEM images at various magnifications were obtained to examine the shape and dimension of the produced submicron structures. Figure 5 reveals the morphology of the samples (a) pure ZnO, (b) ZnO:Ag$_{(2\%)}$, (c) ZnO:Ag$_{(4\%)}$, (d) ZnO:Ag$_{(6\%)}$, (e) ZnO:Ag$_{(8\%)}$, and (f) ZnO:Ag$_{(10\%)}$ examined by SEM. Pure ZnO was formed as homogeneous submicron hexagonal particles. Figure 5a,b show that an array of well-aligned ZnO submicron structures at high density was uniformly grown. Doping-induced morphology changes produced unique submicron structures for Ag-doped ZnO samples. The doping results indicate that Ag doping remarkably impacts ZnO morphology. The surface of ZnO:Ag$_{(2\%)}$ submicron powder was observed as uniformly agglomerated rods-like of analogous frugally bundled together. A further change to an interlocked flower-like morphology was visible in the SEM micrographs of ZnO:Ag$_{(10\%)}$. Figure 5c–f shows a flower comprising hexagonal rod-like crystals with petals emerging from the center. The changes in the sample's morphology can be attributed to the growth of nanostructures differing in the preferred axis. The highest peak for the nanorods was along the (002) direction (Figure 4, pure ZnO), followed by the SEM analyses.

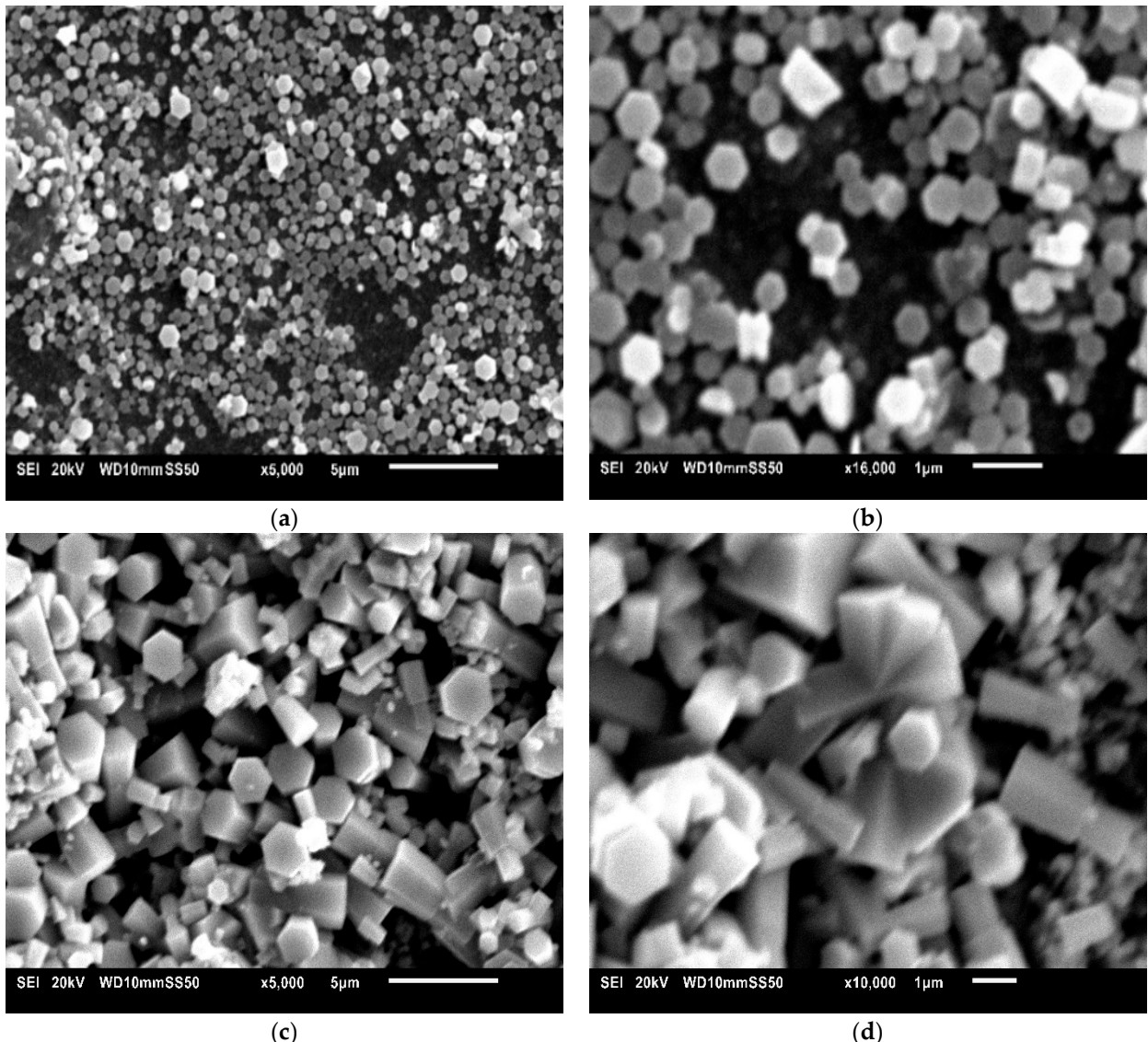

(a)

(b)

(c)

(d)

**Figure 5.** *Cont.*

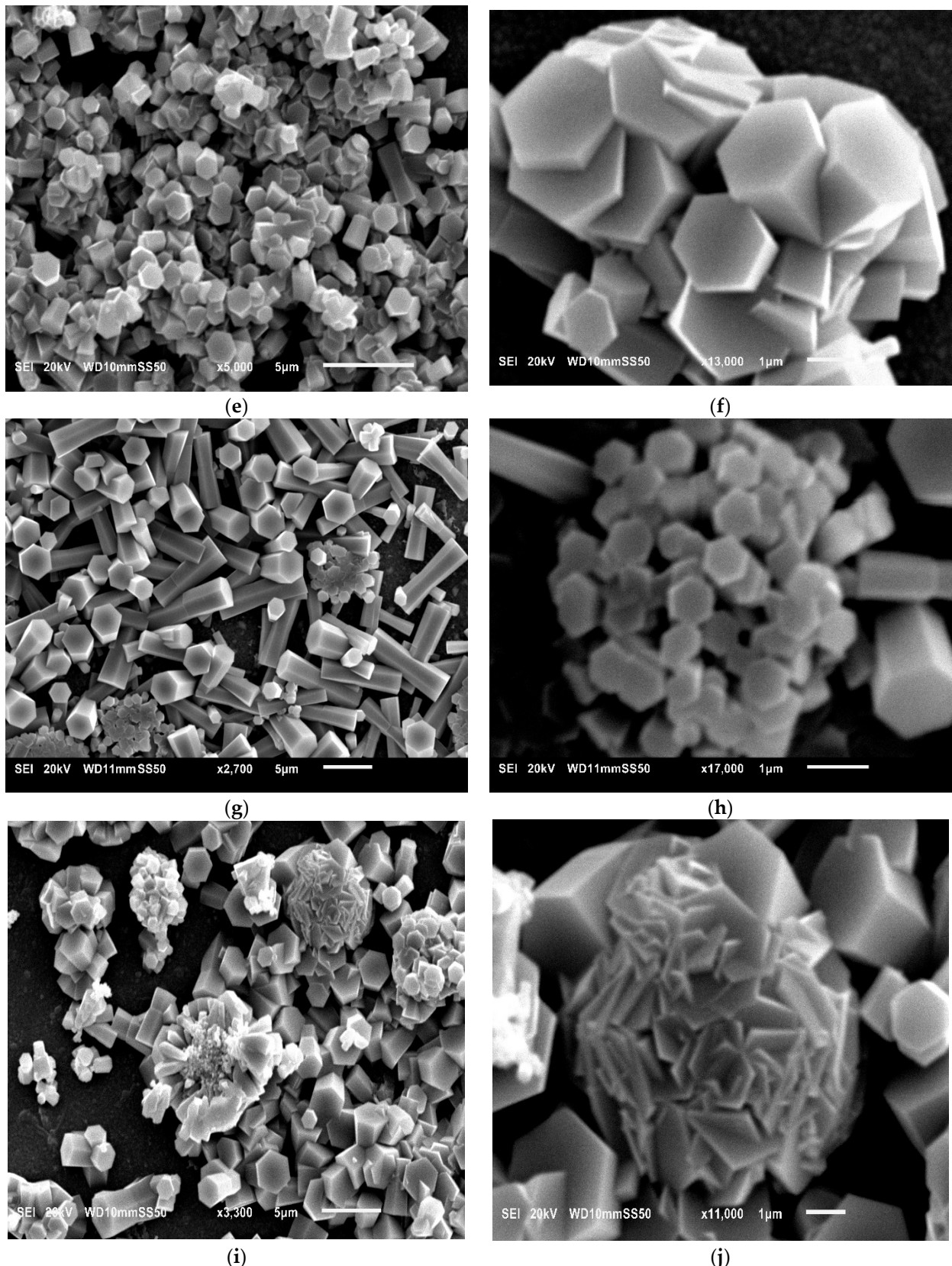

**Figure 5.** *Cont.*

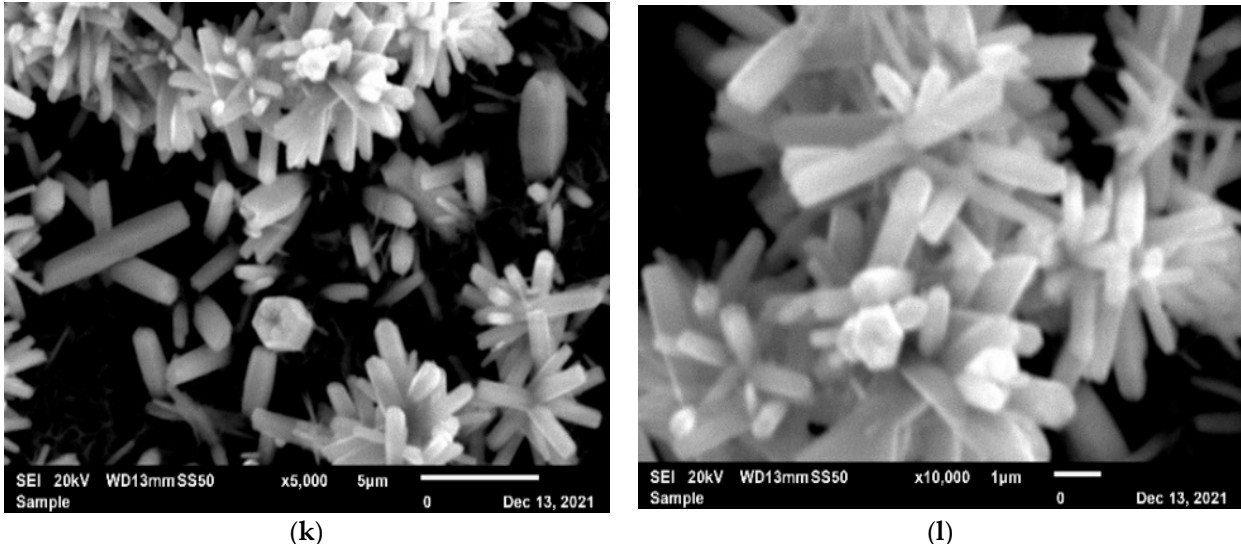

**(k)**                                                                 **(l)**

**Figure 5.** Scanning Electron Microscope (SEM) images of samples (**a,b**) Pure ZnO, (**c,d**) ZnO:Ag$_{(2\%)}$, (**e,f**) ZnO:Ag$_{(4\%)}$, (**g,h**) ZnO:Ag$_{(6\%)}$, (**i,j**) ZnO:Ag$_{(8\%)}$, and (**k,l**) ZnO:Ag$_{(10\%)}$ synthesized by LACBS.

On the contrary, the prominent peaks for submicron structures were (100) and (101), which was consistent with SEM images and confirmed the non-vertical development orientation of nanoflowers. The distributed 3D flower-like structure had a higher surface-to-volume ratio than the 1D structure of a rod-like submicron structure, and this spread shape could capture extra light for improved sensitivity [9]. Due to the increased surface area of the rod and flowers' submicron structure, these structures are crucial for photocatalysts. We may conclude that the morphology of the produced nanoparticles was regulated by the laser irradiation procedure and several other factors. These variables may include the surroundings, the kind of liquid, the laser beam's spot size, and the sort of target. The total surface area was inversely related to the radius of the nanoparticles, and hence the smaller nanosphere had a higher surface area than the bigger one. However, the nanorods had a higher aspect ratio and a bigger surface-to-volume ratio than the submicron spheres [30,31].

*3.3. Energy Dispersive X-ray Diffractive (EDX)*

EDX analysis was conducted for the composition analysis of the fabricated submicron structures. Figure 6 represents the EDX elemental profile of samples (a) pure ZnO, (b) ZnO:Ag$_{(2\%)}$, (c) ZnO:Ag$_{(4\%)}$, (d) ZnO:Ag$_{(6\%)}$, (e) ZnO:Ag$_{(8\%)}$, and (f) ZnO:Ag$_{(10\%)}$ synthesized by LACBS. Figure 6 confirms the presence of Zn, O, and Ag elements using the LACBS method. Pure ZnO is quantitatively composed of zinc and oxygen, nearly in a 1:1 elemental ratio with 50.06% and 49.94% devoid of any other elemental impurities like residual carbon. Here, the Zn to oxygen ratio indicates the presence of enough oxygen vacancies in the sample. The elemental profile of the Ag-doped ZnO sample depicts the elemental composition as shown in Table 2. The doped ZnO samples were also devoid of additional impurity and/or residual peaks.

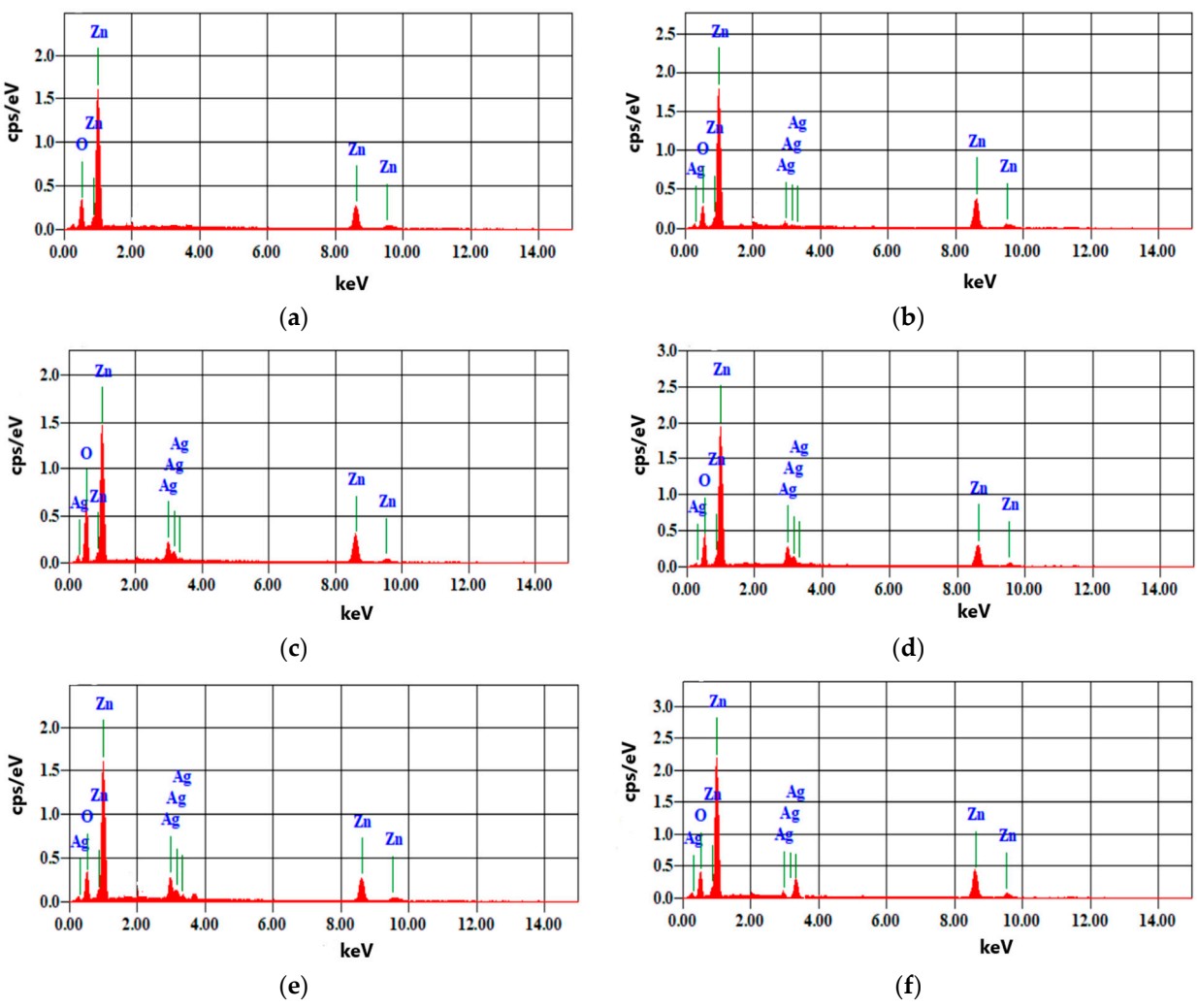

**Figure 6.** EDX spectra of samples (**a**) Pure ZnO, (**b**) ZnO:Ag$_{(2\%)}$, (**c**) ZnO:Ag$_{(4\%)}$, (**d**) ZnO:Ag$_{(6\%)}$, (**e**) ZnO:Ag$_{(8\%)}$, and (**f**) ZnO:Ag$_{(10\%)}$.

**Table 2.** EDX analysis of pure and Ag-doped ZnO nanoparticles.

| Samples | Pure ZnO | | ZnO:Ag$_{(2\%)}$ | | ZnO:Ag$_{(4\%)}$ | | ZnO:Ag$_{(6\%)}$ | | ZnO:Ag$_{(8\%)}$ | | ZnO:Ag$_{(10\%)}$ | |
|---|---|---|---|---|---|---|---|---|---|---|---|---|
| **Elements** | **wt%** | **At%** | **wt%** | **At%** | **wt%** | **At%** | **wt%** | **At%** | **wt%** | **At%** | **wt%** | **At%** |
| O | 19.70 | 50.06 | 10.31 | 32.36 | 14.51 | 42.03 | 24.20 | 58.84 | 19.32 | 52.02 | 15.53 | 45.5 |
| Zn | 80.30 | 49.94 | 85.56 | 65.72 | 76.03 | 53.90 | 58.97 | 35.09 | 60.63 | 39.97 | 62.98 | 45.16 |
| Ag | 0 | 0 | 4.13 | 1.92 | 9.46 | 4.07 | 16.83 | 6.07 | 20.06 | 8.01 | 21.49 | 9.34 |
| Total | 100 | 100 | 100 | 100 | 100 | 100 | 100 | 100 | 100 | 100 | 100 | 100 |

### 3.4. UV-Vis Absorption

UV-Vis absorption spectra were used to analyze the optical band gap of the synthesized sample. Figure 7a represents the absorption spectra of the prepared samples of pure ZnO, ZnO:Ag$_{(2\%)}$, ZnO:Ag$_{(4\%)}$, ZnO:Ag$_{(6\%)}$, ZnO:Ag$_{(8\%)}$, and ZnO:Ag$_{(10\%)}$ synthesized by LACBS. Undoped ZnO exhibits an absorption edge near 389 nm, related to the typical electron transition from the O$^{2p}$ orbital to the Zn$^{3d}$ orbital. The primary electron transitions of ZnO from the valence band to the conduction band were responsible for the shoulder, which denotes the highest absorption peak. After Ag doping, there was a remarkable shift in the absorption edge toward higher wavelengths, which caused the band gap to narrow.

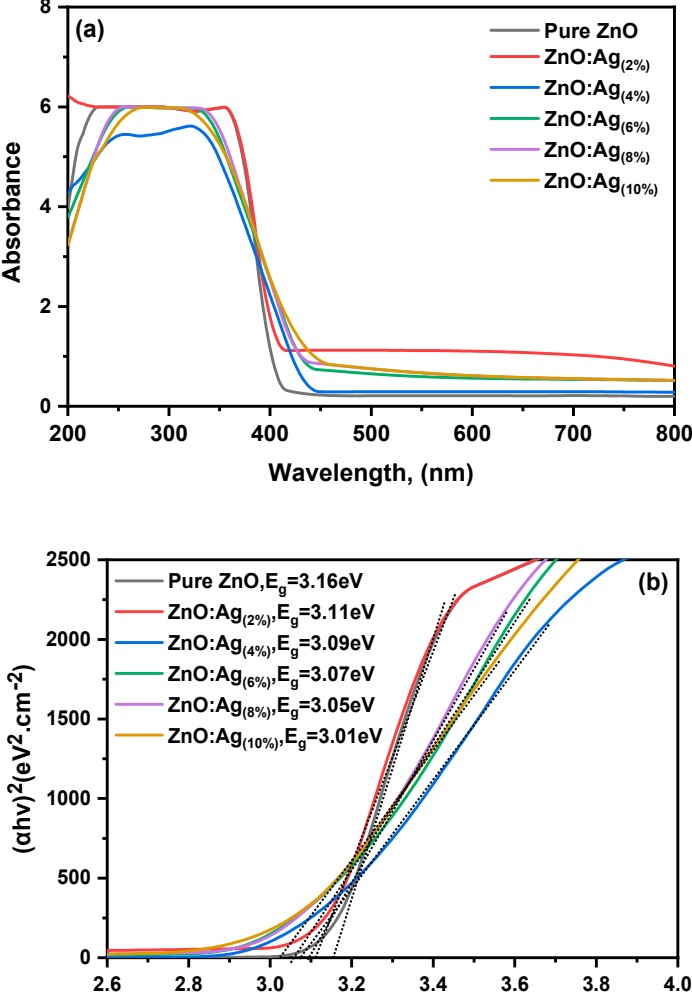

**Figure 7.** (**a**) Absorbance spectra, and (**b**) Optical band gap of pure and Ag-doped ZnO submicron structures.

The presence of Ag in the ZnO matrix caused an absorption edge shift, resulting in a narrower optical band gap with Ag doping in ZnO [32]. The highest absorption for the sample ZnO:Ag$_{(10\%)}$ was about 438.1 nm. This is because the sample, ZnO:Ag$_{(10\%)}$ had a morphology defined by small nanostructures and low porosity.

The optical band gaps of the samples were determined from the UV-visible absorption spectra using the following equation:

$$E_g(eV) = \frac{hc}{\lambda(nm)} = \frac{1240}{\lambda(nm)} \tag{13}$$

where $E_g$ is the optical band gap, $h$ is Planck's constant, $c$ is the speed of light, and $\lambda$ is the maximum absorption wavelength. In addition, optical band gaps of the samples were determined using the Tauc equation (Equation 14), as shown in Figure 7b.

$$(\alpha h\nu)^n = A(h\nu - E_g) \tag{14}$$

where $\alpha$ denotes the absorption coefficient, $A$ the absorption constant, and $n$ is a dimensionless constant depending on the semiconductor type. Thus, the optical band gaps of the samples are shown in Figure 7. The contraction in the optical band gap can be attributed to the crystal defects and quantum confinement induced by the dopants. The variability in band gap values with Ag doping of ZnO (Figure 7b) may be related to the availability

of oxygen vacancies, which easily transport electrons from the valence band (VB) to the conduction band (CB) [33]. In other words, when the concentration of oxygen vacancies grows, the impurity states of ZnO become increasingly delocalized and overlap with the valance band edge, lowering the ultimate band gap.

### 3.5. FTIR

Typically, the peaks between 400 and 600 $cm^{-1}$ are attributed to the stretching mode of metal-oxygen (M-O). According to studies in the literature, peaks at around 437–455 $cm^{-1}$ provide information about the Zn-O stretching vibration of crystalline hexagonal zinc oxide [34]. As seen in Figure 8, the characteristic ZnO band can be seen in the spectra of both undoped and Ag-doped ZnO nanostructures at a wavelength of about 530 $cm^{-1}$. The O-H stretching of the hydroxyl group or water absorbed on the surface of the nanoparticles caused the band at about 3300–3700 $cm^{-1}$. No silver bands can be seen in the silver-modified ZnO, proving no chemical link between silver and ZnO. The peak at 1500 is associated with bending vibrations of $Zn(OH)_2$ and stretching modes of vibrations in both symmetric and asymmetric C=O bonds. Additionally, some bands developed during the creation of nanoparticles from airborne carbon dioxide and water moisture. Due to the absorption of ambient $CO_2$ on the surface of metal cations, a peak was seen at about 2350 $cm^{-1}$ [35].

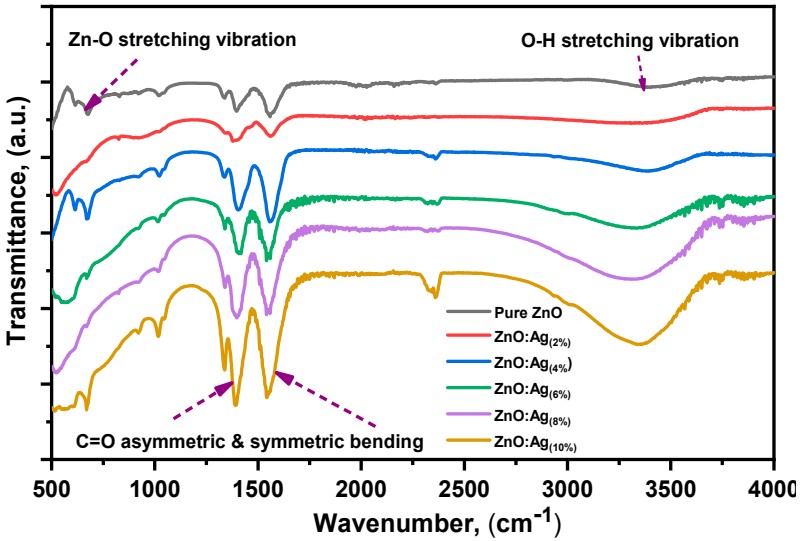

**Figure 8.** FTIR spectrum of pure and Ag-doped ZnO submicron structures.

### 3.6. Photocatalyst Study

3.6.1. Photocatalytic Degradation under Blue Laser Irradiation/UV-Vis Studies

In the presence of blue laser light ($\lambda$ = 444.5 nm and I = 8000 lx), and all experiments were carried out simultaneously, the prepared pure and Ag-doped ZnO submicron structures could be used as a photocatalyst to convert methylene blue (MB) dye to colorless form. The ability of the pure and Ag-doped ZnO submicron structure photocatalysts to reduce MB dye when exposed to blue laser radiation is shown in Figure 9a–f. A UV-Vis spectrophotometer measured the reaction between 200 and 800 nm at room temperature. The MB solution's UV-Vis absorbance spectrum's peak absorption always appeared at 656.1 nm. The absorption peak decreased as exposure time increased in the presence of a photocatalyst and blue laser light. Compared to pure ZnO, Ag-doped ZnO submicron structures exhibited a faster decrease in absorbance peak. The MB solution's bright blue intermittently faded and turned colorless as it degraded. The MB spectra were systematically lowered with time, which showed MB degradation and good photocatalyst action. Using ZnO:Ag (10%) flower-like submicron structure as a catalyst caused further degradation because of the increased active surfaces and large surface area. This behavior occurred

because additional hydroxyl radicals were produced and moved to the surface as electron pairs, increasing the quantity and catalytic efficiency [36]. The findings suggest employing less catalyst in the next operations when longer timeframes are permitted. To fully understand the consequences of using laser irradiation for manufacturing ZnO nanoparticles for photocatalyst degradation, more research must be done by carefully examining the environments created throughout the reactions. Future research should investigate additional salts, such as zinc nitrate, and different liquids, such as methanol, acetone, and benzyl alcohol. Future research on photocatalysts still must consider the effects of other variables, including temperature, pH, light intensity, catalyst dose, and dye concentration.

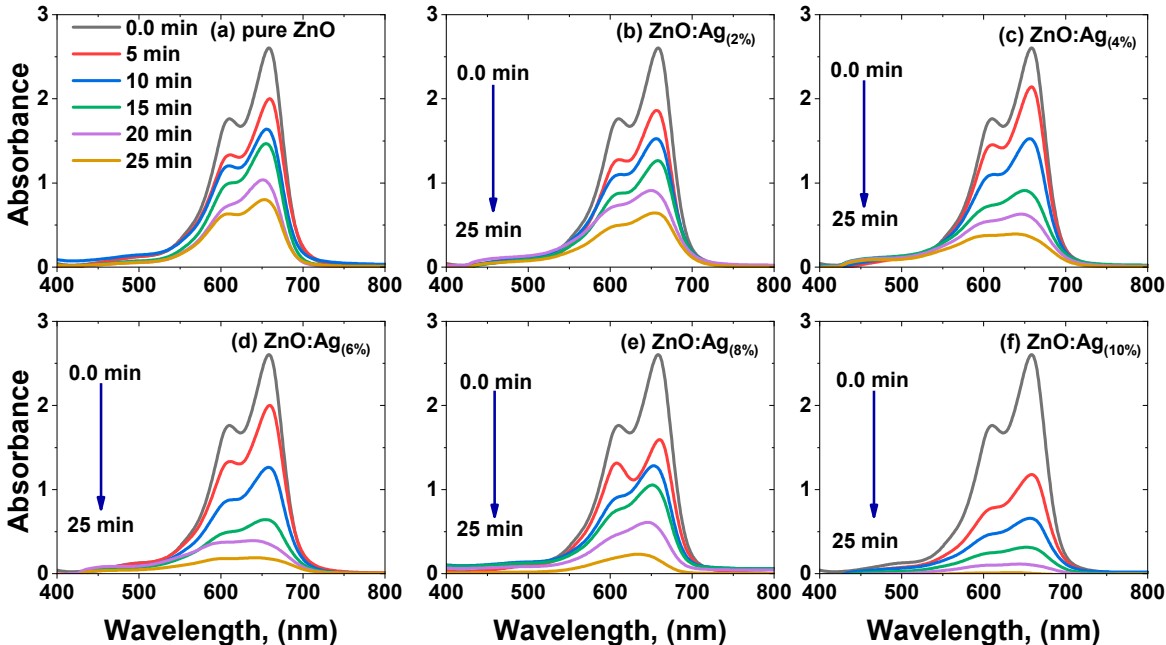

**Figure 9.** UV-Visible absorption spectra of MB dye using (**a**) pure ZnO, (**b**) ZnO:Ag$_{(2\%)}$, (**c**) ZnO:Ag$_{(4\%)}$, (**d**) ZnO:Ag$_{(6\%)}$, (**e**) ZnO:Ag$_{(8\%)}$, and (**f**) ZnO:Ag$_{(10\%)}$.

The photodegradation efficiency (PDE%) was calculated according to Equation (3) as a quantitative representation of the degraded dye, as shown in Figure 10. The photocatalytic degradation was found to increase linearly with increasing irradiation time. After 5 min of stirring the dye mixture under blue laser irradiation, the recorded adsorption of pure ZnO (23.2 %), ZnO:Ag$_{(2\%)}$ (29.6%), ZnO:Ag$_{(4\%)}$ (19.1 %), ZnO:Ag$_{(6\%)}$ (26.2 %), ZnO:Ag$_{(8\%)}$ (40.1%), and ZnO:Ag$_{(10\%)}$ (55.4 %) in the dye solution were increased to 68.5%, 74.9%, 83.5%, 92.9%, 94.1%, and 99.3%, respectively, after 25 min of irradiation. It demonstrated that the higher the doping concentration, the greater the degree of enhancement in the case of ZnO:Ag$_{(10\%)}$ adsorption. The degradation process may be significantly caused by the Ag-doped ZnO catalyst's low band gap energy. Ag doping dramatically improved photocatalytic activity by increasing surface area and oxygen defects. The short-term increase in activity of Ag-doped ZnO submicron structures makes them suitable for cost-friendly and environmentally friendly pollutant remediation [37].

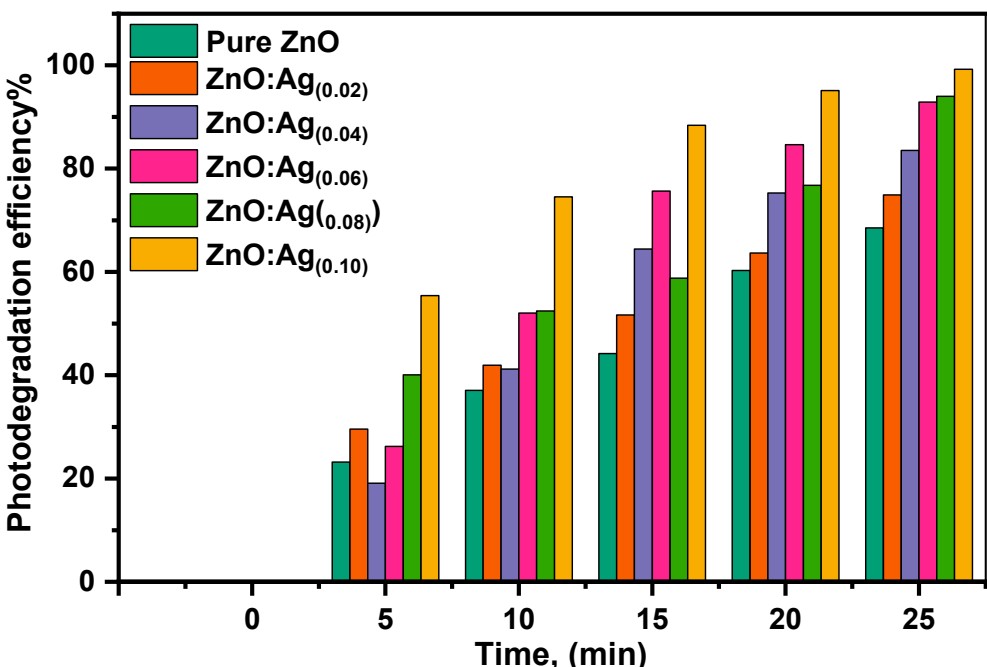

**Figure 10.** Percentage of MB dye degradation at different times and varying photocatalysts.

The rate of dye degradation was calculated using photocatalytic kinetics. Figure 11 shows the plot of $C/C_o$ against time

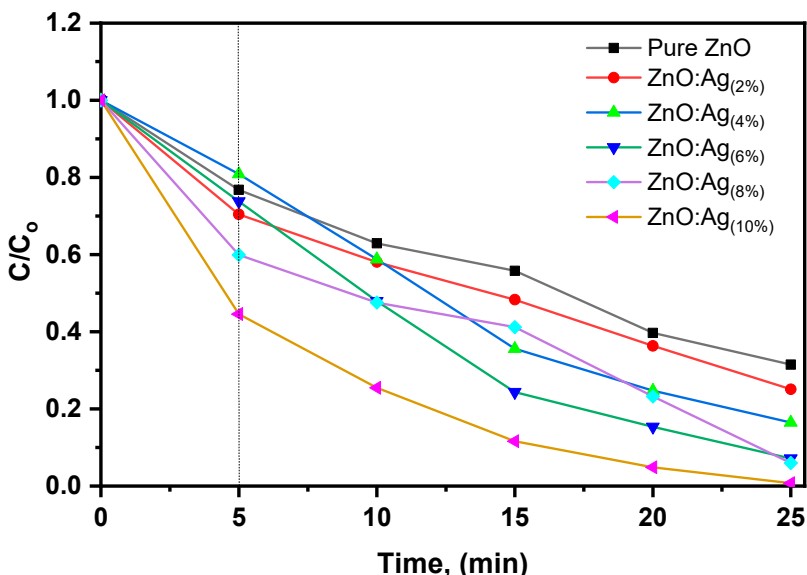

**Figure 11.** Concentration spectra of MB using pure and Ag-doped ZnO submicron structures as a catalyst.

Figure 12 shows the linear plots of $\ln(C/C_o)$ for the photodegradation of MB using pure and Ag-doped ZnO as photocatalysts under the blue laser. $C_O$ and $C$ is the concentration of MB when the reaction time is 0 and t, respectively. The decolorization and degradation of MB are shown to be consistent with pseudo-first-order kinetics in this figure. According to Equation (2), the slopes of the plots that reflect the photocatalyst reaction rate constant were determined and are shown in Table 3. The rate constants differed significantly from those previously reported. The figure shows that the photocatalyst activity was highest in nanorods and nanoflowers. This is because these structures can transfer elec-

trons more quickly than other structures. This results from quantum confinement, which enhances just one free transportation path. In addition, the efficiency of photocatalysts is frequently greatly influenced by surface flaws. As previously stated, vacancies are the most common crystallographic defects in ZnO [36]. It is believed that the ZnO principally has exposed an active $O^{2-}$ on (101), (002), and (110) facets. Therefore, these facets are deliberated as an active center for degradation. Recent studies have examined how ZnO morphologies affect their photocatalytic activity. According to Zhang et al., raising the aspect ratio of ZnO flower-like could lead to more surface defects and higher photocatalytic efficacy [38]. According to Han et al., the chemisorption capacity of the reacted surfaces regulates the photocatalytic activity of ZnO nanostructures [39].

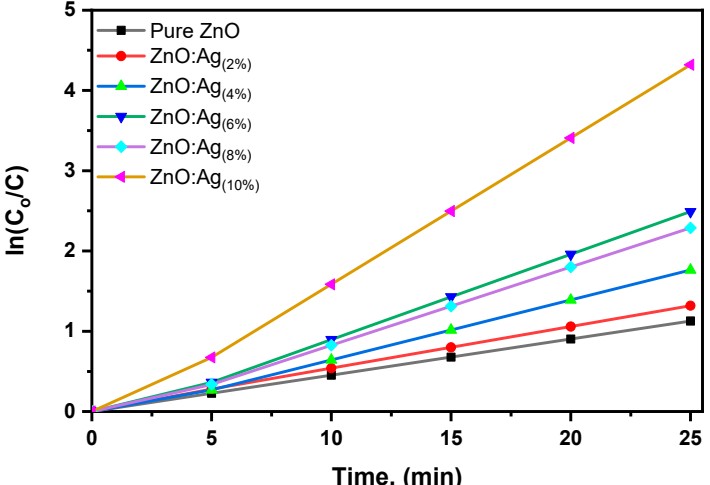

**Figure 12.** $\ln\left(\frac{C_o}{C}\right)$ versus time plot to determine the rate of constant.

**Table 3.** Comparison between efficiencies of different catalyst systems with previous studies.

| Synthesis Method | Catalyst Type | Catalyst Amount | Dye Concentration | Light Source | %Deg | $k_{app}$ (min$^{-1}$) | Time (min) | Ref. |
|---|---|---|---|---|---|---|---|---|
| Co-precipitation | ZnO:Ag$_{(0.05)}$ | 10 mg/100 mL | MB (20 ppm) | visible | 98 | 0.017 | 120 | [40] |
| Co-precipitation | ZnO:Ag$_{(0.10)}$ | 15 mg/25 mL | MB (25 ppm) | visible | 98 | | 120 | [41] |
| Hydrothermal | ZnO:Ag$_{(0.05)}$ | 50 mg/100 mL | MB (10 ppm) | visible | 92.9 | | 210 | [42] |
| Co-precipitation | ZnO:Ag$_{(0.02)}$ | 10 mg/50mL | MB (20 ppm) | visible | 96 | | 80 | [43] |
| LACBS | ZnO | 10 mg/50 mL | MB (20 ppm) | Blue | 68.5 | 0.045 | 25 | This work |
| LACBS | ZnO:Ag$_{(2\%)}$ | 10 mg/50 mL | MB (20 ppm) | Blue | 68.5 | 0.052 | 25 | This work |
| LACBS | ZnO:Ag$_{(4\%)}$ | 10 mg/50 mL | MB (20 ppm) | Blue | 83.5 | 0.097 | 25 | This work |
| LACBS | ZnO:Ag$_{(6\%)}$ | 10 mg/50 mL | MB (20 ppm) | Blue | 92.9 | 0.106 | 25 | This work |
| LACBS | ZnO:Ag$_{(8\%)}$ | 10 mg/50 mL | MB (20 ppm) | Blue | 94.1 | 0.098 | 25 | This work |
| LACBS | ZnO:Ag$_{(10\%)}$ | 10 mg/50 mL | MB (20 ppm) | Blue | 99.3 | 0.183 | 25 | This work |

Numerous studies have been done on pure and Ag-doped ZnO nano-photocatalysts for degrading organic dyes. We contrast this novel pure and Ag-doped ZnO with the earlier studies in Table 3. Pure ZnO has the maximum efficiency compared to earlier works, as shown in Table 3. Furthermore, the highest efficiency for MB degradation was observed with the most significant amount of Ag-doped in ZnO$_{(10\%)}$

### 3.6.2. Photocatalytic Stability

The photostability of a photocatalyst is a critical criterion for practical application. The process of photocatalysis accelerates photoreaction in the catalyst's presence. Generally, photocatalytic activity relies on the capacity to generate electron-hole pairs ($e^- - h^+$), which cause free radicals like $O_2$ and ($\cdot OH$). Five photocatalytic experiments were conducted using old photocatalysts in new MB solutions with no change in the catalyst's overall concentration when exposed to blue laser for the best sample [ZnO:Ag$_{(10\%)}$]. After each recycles, the photocatalyst was centrifuged, rinsed with double-distilled water, and dried at 85 °C. However, the other parameters were left unchanged. As a result, the photocatalyst may be stable. The photodegradation efficiency (PDE%) was calculated according to Equation (3) as a quantitative representation of the degraded dye, as shown in Figure 13a. The photocatalytic degradation was found to decrease slightly with the repetition of the experiment. After 25 min of stirring the dye mixture under blue laser irradiation, the recorded adsorption was as follows: cycle 1 (99.30%), cycle 2 (98.85%), cycle 3 (97.36%), cycle 4 (96.78%), and cycle 5 (95.46%). The photocatalytic activity of the photocatalyst, which remained in the range of 95.5% after five consecutive testing runs, revealed that the photocatalytic activity of the photocatalyst has great potential for use in water purification. Figure 13b shows the rate of degradation for the photodegradation of MB for five cycles of catalyst [ZnO:Ag$_{(10\%)}$] as photocatalysts under the blue laser. According to Equation (2), the slopes of the plots that reflect the photocatalyst reaction rate constant were determined and recorded in cycle 1 (0.183 min$^{-1}$), cycle 2 (0.176 min$^{-1}$), cycle 3 (0.164 min$^{-1}$), cycle 4 (0.157 min$^{-1}$), and cycle 5 (0.142 min$^{-1}$).

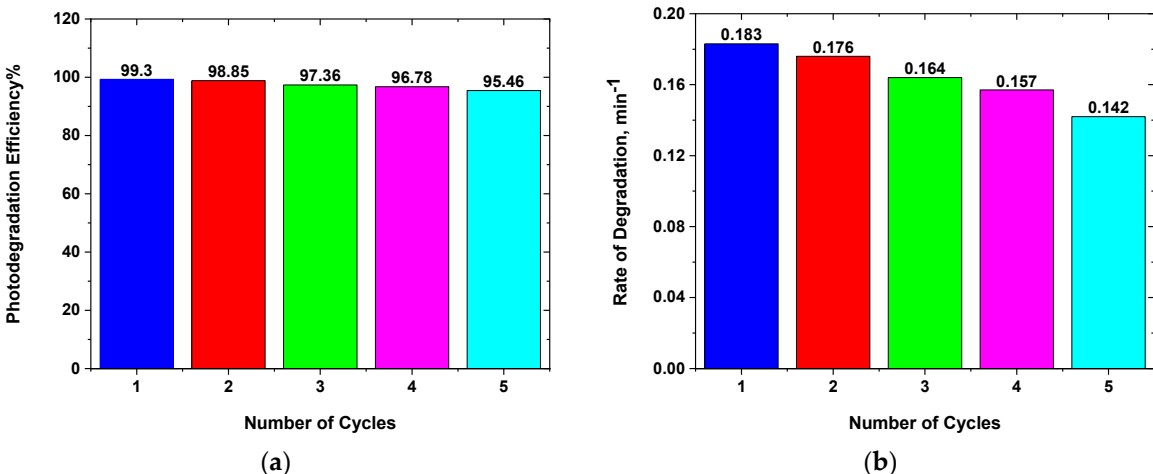

**Figure 13.** (**a**) Photodegradation efficiency with a number of cycles, (**b**) Rate of degradation with number of cycles, for recycling the catalyst [ZnO:Ag$_{(10\%)}$].

As displayed in Figure 14, the XRD pattern of the catalyst sample [ZnO:Ag$_{(10\%)}$] shows phase stability after five cycles, suggesting the chemical structure and composition of the photocatalyst were hardly influenced throughout the degradation process. The results showed that the prepared sample was a stable and recyclable photocatalytic material for MB dye removal from water.

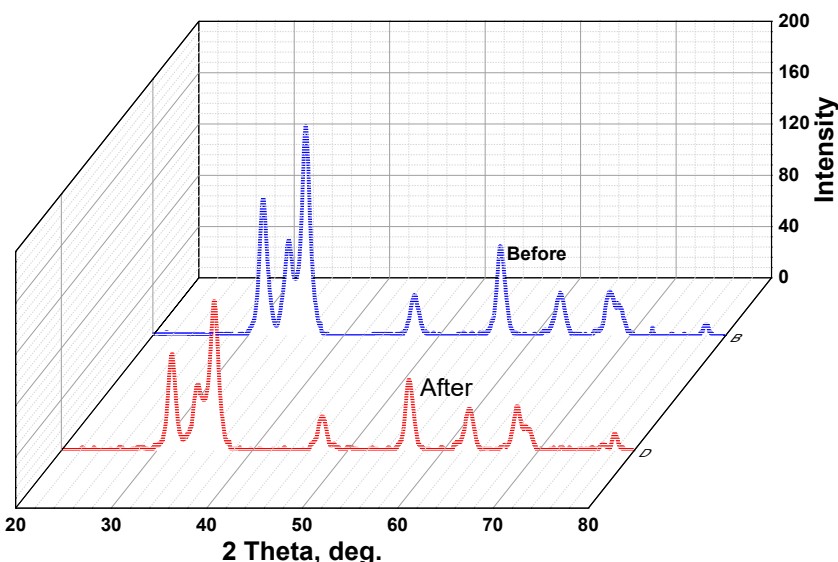

**Figure 14.** XRD patterns of catalyst [ZnO:Ag$_{(10\%)}$] before (Cycle 1) and after (Cycle 5) stability tests.

### 3.6.3. The Mechanism of Photocatalyst

Figure 15 shows a schematic of the heterogeneous photocatalysis mechanism. A wide band gap ZnO catalyst is included. While the hole drifts to the bottom of the valence band, the photo-generated electron goes to the conduction band. While some of these photo-generated charge carriers escape recombination and begin redox reactions in molecules adsorbed on the surface of the photocatalyst, destroying them, most of these charge carriers undergo wasteful recombination [44]. Due to their high redox potentials, photo-generated electrons and holes have been discovered to destroy practically all forms of organic, inorganic, and microbiological pollutants, improving photocatalytic performance by noble metal deposition on semiconductor nanoparticles [45]. The metal modifier may indirectly affect the charge transfer mechanisms at the interface. Silver is a noble metal that can capture and efficiently store electrons [40]. To properly define the precise function of metal in semiconductor photocatalysis, a basic understanding of photo-induced interactions and interfacial charge transfer mechanisms in metal-modified semiconductors is required. Basic knowledge of photo-induced interactions and interfacial charge transfer mechanisms in metal-modified semiconductors are needed to determine the exact function of metal in semiconductor photocatalysis properly. Figure 15b depicts the potential catalytic mechanism of Ag-doped ZnO. The valence band electron in ZnO will be pushed into the conduction band during the catalytic process, producing an equal amount of valence band holes [41].

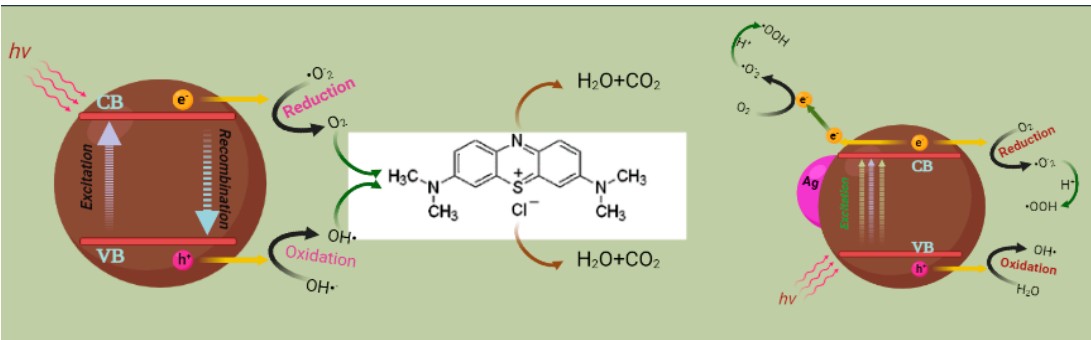

**Figure 15.** Schematic diagram of the proposed mechanism for the degradation of MB by pure ZnO, and Ag-doped ZnO nanostructures.

Meanwhile, photoexcited electrons would eagerly move from ZnO to Ag submicron structures under the potential energy due to the lower Fermi level (Ag-doped ZnO) compared to the conduction band energy of ZnO. This would cause photo-generated electrons to accumulate in Ag nanoparticles and photo-generated holes to move to the surface of ZnO, which significantly aids in the separation of photo-generated electrons and holes [42]. Separation and recombination of photo-induced charge carriers are competitive paths in such a photocatalytic process, and photocatalytic activity is successful when recombination between them is inhibited. Afterward, as a receptor, photoelectrons will adsorb $O_2$ and turn to superoxide anion ($\cdot O^{-2}$), the obtained ($\cdot O^{-2}$) and photo-generated holes would finally turn into a strong oxidizing agent ($\cdot OH$). It can successfully convert organic dye to $CO_2$ and water, greatly enhancing photocatalytic performance [43,46,47]. The following is an expression for the potential photocatalytic reaction:

$$ZnO + h\nu \rightarrow ZnO\left(e_{CB}^- + h_{VB}^+\right) \tag{15}$$

$$e_{CB}^- + O \rightarrow \cdot O^{-2} \tag{16}$$

$$Ag^+ + e_{CB}^- \rightarrow Ag \tag{17}$$

$$h_{VB}^+ + OH^- \rightarrow \cdot OH \tag{18}$$

$$O^{-2} + 2H^+ \rightarrow 2\cdot OH \tag{19}$$

$$H + MB \rightarrow Degradation\ Products \tag{20}$$

## 4. Conclusions

In the present study, a novel method named LACBS has been used to prepare pure and Ag-doped ZnO submicron structures. Due to its low cost, simplicity, high production, and outstanding performance of the final products, this approach would be required for using ZnO photocatalytic materials. Photodegradation has also been studied under blue laser irradiation with 444.5 nm of wavelength for the first time. In addition, the effect of Ag doping concentration has been investigated. XRD, FESEM, EDS, UV-Vis, and FTIR were used to evaluate the structural and optical characteristics of the resulting materials. The photocatalytic activity of the prepared pure and Ag-doped ZnO submicron structures was assessed by tracking the degradation of methylene blue. The interfacial charge transfer processes are made more accessible by silver's presence, according to research on the mechanism of photocatalytic activity. The findings demonstrate silver's significant contribution to the electron-trapping properties of these materials. Additionally, oxygen defects are essential for improving photocatalytic efficiency. The development of a photocatalyst usable in visible regions is thus a potential extension of the applications of these materials.

**Author Contributions:** Conceptualization, S.H.Z. and A.H.Z.; methodology, S.H.Z. and I.S.Y.; software, M.G.D. and G.N.M.; validation, S.H.Z.; formal analysis, S.H.Z. and H.Y.Z.; investigation, M.N.; resources, M.S. and N.Q.; data curation, S.H.Z. and A.A.; writing—original draft preparation, S.H.Z., G.N.M. and M.S.A.-w.; writing—review and editing, S.H.Z., M.S.A.-w. and I.S.Y.; visualization, N.H.; supervision, S.H.Z.; project administration, S.H.Z.; funding acquisition, S.H.Z. All authors have read and agreed to the published version of the manuscript.

**Funding:** This research was funded by the Deanship of Graduate Studies and Research, Ajman University grant number DGSR Ref. Number: 2022-IRG-HBS-17 and The APC was funded by Ajman University.

**Institutional Review Board Statement:** Not applicable.

**Informed Consent Statement:** Not applicable.

**Data Availability Statement:** Data are contained within the article.

**Conflicts of Interest:** The authors declare no conflict of interest.

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
