# Peer review of "Fast and Excellent Enhanced Photocatalytic Degradation of Methylene Blue Using Silver-Doped Zinc Oxide Submicron Structures under Blue Laser Irradiation"

_crystals, doi:10.3390/cryst13020229_

Round 1

Reviewer 1 Report

Comments:

0. Major revision. 1. The novelty of this study should be inserted in the text clearly. 2. The advantages and disadvantages of the photocatalysts should be investigated. 3.  The regeneration of the photocatalysts should be presented in detail. 4. The stability of the photocatalysts after the degradation process should be presented by XRD. 5. The ion leaching from the photocatalysts during the degradation process should be indicated. 6. The “introduction” and “results and discussion” sections of the manuscript can be strengthened and supported with some papers related to the literature and cited (optional for authors): Journal of Molecular Liquids 282 (2019), 115-130; Biotechnology and Bioprocess Engineering 20 (2015), 109-116

Author Response

Dear reviewer,

On the behalf of my co-authors, I would like to thank all of you for careful and thorough reading our manuscript and for the helpful comments and constructive and suggestions, to improve the manuscript.  Our response follows your comments sequences. We hope that you find our response satisfactory, and that manuscript in now acceptable for publication in respect Crystals.

Sincerely yours,

On the behalf of co-authors

Reviewer 2 Report

Dear authors, please review the manuscript crystals-2144337 to improve it.

In Title: …Using Silver-Doped Zinc Oxide Nanostructures … sorry I see microstructures or submicron structures - Change it in the title

In Abstract: rewrite the paragraphs:

remove In this project, our novelty…;

rewrite (l = 444.5 nm, I = 8000 lx)…;

rewrite  … The doping concentration varied from 2%, 4%, 6%, 8%, and 10% …;

rewrite … continuous blue laser (P = 7 W, l = 444.5 nm)…

In Abstract:

rewriteDue to a combination of the Ag plasmonic effect and ZnO surface imperfections that facilitate the separation of photo-generated electron-hole pairs and shift the absorption edge of the hybrid nanostructure toward the visible spectrum region, the photocatalytic activity is increased …

In introduction:

Included references (Check throughout the introduction)

Chemicals with a carbon base and persistent organic pollutants are resistant to environmental deterioration and may not be eliminated by treatment procedures …

the solar photocatalysis process has drawn increasing interest.

… organic contaminants … Which?

In methodology:

Use min by minutes, h by hours, g by grams, °C correct it in the document – check in all document

Describe using paragraphs the preparation of the samples (rewrite section 2.2)  

In section 2.3 change (=444.4 nm and P=7 W) by laser (P = 7 W, l = 444.5 nm)…

2%, … 10 %  is wt % explain it in the manuscript

In section 2.6.2 check the paragraph and review:…  In order to eliminate any thermal catalytic influence, the photocatalytic degradation of MB was studied at room temperature under blue laser light irradiation(P = 7 W, l = 444.5 nm)

Use the same code i.e., ZnO:Ag(0.02) or ZnO:Ag(2%).In all manuscript including the figures and captions

Results and Discussion

(444.5 nm of wavelength and 7 W of output power) use (P = 7 W, l = 444.5 nm)…

Table 1. Use significant figures/Significant numbers

Always keep the least number of significant figures

Table 1 is chaotic/confusing: use uncertainty which is the least you can measure with the equipment used

In figure 5, the scale bar is not visible, you should improve these figures

Remember, ., ZnO:Ag(0.02) or ZnO:Ag(2%). Check Tables , figures/ captions and the manuscript

Nanostructures... not clear... less than 100 nm is considered nano... could be considered submicron structures

In results review the follow manuscript https://doi.org/10.1016/j.apsusc.2015.04.148 and discuss the results respect to surface plasmon resonance, and local vibration modes due to silver incorporation in ZnO. Complete the manuscript with Raman spectroscopy if possible.

Conclusion

review the conclusions and rewrite it with the most relevant of your work including the main results

Review the entire manuscript and keep the same nomenclature, units, etc. Check the English, and check that all authors have their correct affiliations.

Author Response

(The authors gave the same response as above.)

Round 2

Reviewer 1 Report

Accept

Author Response

Dear Reviewer

We appreciate your decision, Thanks

Reviewer 2 Report

Dear authors,

Is required to change the scale bars of Figure 5 in all images. After that, the manuscript will be accepted for publishing

Author Response

Dear reviewer 

On the behalf of my co-authors, I would like to thank all of you for careful and thorough reading our manuscript and for the helpful comments and constructive and suggestions, to improve the manuscript.  Our response follows your comments sequences. We hope that you find our response satisfactory, and that manuscript in now acceptable for publication in respect Crystals.

Figure 5 has been modified (please see in manuscript)

Sincerely yours,

On the behalf of co-authors
